# Mapping State Space using Landmarks for Universal Goal Reaching

**Zhiao Huang** *
UC San Diego
z2huang@eng.ucsd.edu

**Fangchen Liu** *
UC San Diego
fliu@eng.ucsd.edu

**Hao Su**
UC San Diego
haosu@eng.ucsd.edu

## Abstract

An agent that has well understood the environment should be able to apply its skills for any given goals, leading to the fundamental problem of learning the Universal Value Function Approximator (UVFA). A UVFA learns to predict the cumulative rewards between all state-goal pairs. However, empirically, the value function for long-range goals is always hard to estimate and may consequently result in failed policy. This has presented challenges to the learning process and the capability of neural networks. We propose a method to address this issue in large MDPs with sparse rewards, in which exploration and routing across remote states are both extremely challenging. Our method explicitly models the environment in a hierarchical manner, with a high-level dynamic landmark-based map abstracting the visited state space, and a low-level value network to derive precise local decisions. We use farthest point sampling to select landmark states from past experience, which has improved exploration compared with simple uniform sampling. Experimentally we showed that our method enables the agent to reach long-range goals at the early training stage, and achieve better performance than standard RL algorithms for a number of challenging tasks.

## 1 Introduction

Reinforcement learning (RL) allows training agents for planning and control tasks by feedbacks from the environment. While significant progress has been made in the standard setting of achieving a goal known at training time, e.g., to reach a given flag as in MountainCar [1], very limited efforts have been exerted on the setting when goals at evaluation are unknown at training time. For example, when a robot walks in an environment, the destination may vary from time to time. Tasks of this kind are unanimous and of crucial importance in practice. We call them Universal Markov Decision Process (UMDP) problems following the convention of [2].

Pioneer work handles UMDP problems by learning a *Universal Value Function Approximator* (UVFA). In particular, Schaul et al. [3] proposed to approximate a goal-conditioned value function $V(s,g)^2$ by a multi-layer perceptron (MLP), and Andrychowicz et al. [4] proposed a framework called *hindsight experience replay* (HER) to smartly reuse past experience to fit the universal value function by TD-loss. However, for complicated policies of long-term horizon, the UVFA learned by networks is often not good enough. This is because UVFA has to memorize the cumulative reward between all the state-goal pairs, which is a daunting job. In fact, the cardinality of state-goal pairs grows by a high-order polynomial over the horizon of goals.

While the general UMDP problem is extremely difficult, we consider a family of UMDP problems whose state space is a low-dimension manifold in the ambient space. Most control problems are

of this type and geometric control theory has been developed in the literature [5]. Our approach is inspired by manifold learning, e.g., Landmark MDS [6]. We abstract the state space as a small-scale map, whose nodes are landmark states selected from the experience replay buffer, and edges connect nearby nodes with weights extracted from the learned local UVFA. A network is still used to fit the local UVFA accurately. The map allows us to run high-level planning using pairwise shortest path algorithm, and the local UVFA network allows us to derive an accurate local decision. For a long-term goal, we first use the local UVFA network to direct to a nearby landmark, then route among landmarks using the map towards the goal, and finally reach the goal from the last landmark using the local UVFA network.

Our method has improved sample efficiency over purely network learned UVFA. There are three main reasons. First, the UVFA estimator in our framework only needs to work well for local value estimation. The network does not need to remember for faraway goals, thus the load is alleviated. Second, for long-range state-goal pairs, the map allows propagating accurate local value estimations in a way that neural networks cannot achieve. Consider the extreme case of having a long-range state-goal pair never experienced before. A network can only guess the value by extrapolation, which is known to be unreliable. Our map, however, can reasonably approximate the value as long as there is a path through landmarks to connect them. Lastly, the map provides a strong exploration ability and can help to obtain rewards significantly earlier, especially in the sparse reward setting. This is because we choose the landmarks from the replay buffer using a farthest-point sampling strategy, which tends to select states that are closer to the boundary of the visited space. In experiments, we compared our methods on several challenging environments and have outperformed baselines.

Our contributions are: First, We propose a sample-based method to map the visited state space using landmarks. Such a graph-like map is a powerful representation of the environment, maintains both local connectivity and global topology. Second, our framework will simultaneously map the visited state space and execute the planning strategy, with the help of a locally accurate value function approximator and the landmark-based map. It is a simple but effective way to improve the estimation accuracy of long-range value functions and induces a successful policy at the early stage of training.

## 2   Related work

Variants of goal-conditioned decision-making problems have been studied in literature [7, 8, 3, 9]. We focus on the goal-reaching task, where the goal is a subset of the state space. The agent receives meaningful rewards if and only if it has reached the goal, which brings significant challenges to existing RL algorithms. A significant recent approach along the line is Hindsight Experience Replay (HER) by Andrychowicz et al [4]. They proposed to relabel the reached states as goals to improve data efficiency. However, they used only a single neural network to represent the $Q$ value, learned by DDPG [10]. This makes it hard to model the long-range distance. Our method overcomes the issue by using a sample-based map to represent the global structure of the environment. The map allows to propagate rewards to distant states more efficiently. It also allows to factorize the decision-making for long action sequences into a high-level planning problem and a low-level control problem.

Model-based reinforcement learning algorithms usually need to learn a local forward model of the environment, and then solve the multi-step planning problem with the learned model [11, 12, 13, 14, 15, 16]. These methods rely on learning an accurate local model and require extra efforts to generalize to the long term horizon [17]. In comparison, we learn a model of environment in a hierarchical manner, by a network-based local model and a graph-based global model (map). Different from previous works to fit forward dynamics in local models, our local model distills local cumulative rewards from environment dynamics. In addition, our global model, as a small graph-based map that abstracts the large state space, supports reward propagation at long range. One can compare our framework with Value Iteration Networks (VIN) [18]. VIN focused on the 2D navigation problem. Given a predefined map of known nodes, edges, and weights, it runs the value iteration algorithm by ingeniously simulating the process through a convolutional neural network [19]. In contrast, we construct the map based upon the learned local model.

Sample-Based Motion Planning (SBMP) has been widely studied in the robotics context [20, 21, 22]. The traditional motion planning algorithm requires the knowledge of the model. Recent work has combined deep learning and deep reinforcement learning for [23, 24, 25, 26]. In particularly, PRM-RL addressed the 2D navigation problem by combining a high-level shortest path-based planner

and a low-level RL algorithm. To connect nearby landmarks, it leveraged a physical engine, which depends on sophisticated domain knowledge and limits its usage to other general RL tasks. In the general RL context, our work shows that one can combine a high-level planner and a learned local model to solve RL problems more efficiently. Some recent work also utilize the graph structure to perform planning [27, 28], however, unlike our approach that discovers the graph structure in the process of achieving goals, both [27, 28] require supervised learning to build the graph. Specifically, [27] need to learn a Siamese network to judge if two states are connected, and [28] need to learn the state-attribute mapping from human annotation.

Our method is also related to hierarchical RL research [2, 29, 30]. The sampled landmark points can be considered as sub-goals. [2, 30] also used HER-like relabeling technique to make the training more efficient. These work attack more general RL problems without assuming much problem structure. Our work differs from previous work in how high-level policy is achieved. In their methods, the agent has to learn the high-level policy as another RL problem. In contrast, we exploit the structure of our universal goal reaching problem and find the high-level policy by solving a pairwise shortest path problem in a small-scale graph, thus more data-efficient.

## 3 Background

*Universal Markov Decision Process* (UMDP) extends an MDP with a set of goals $\mathcal{G}$. UMDP has reward function $\mathcal{R} : \mathcal{S} \times \mathcal{A} \times \mathcal{G} \rightarrow \mathcal{R}$, where $\mathcal{S}$ is the state space and $\mathcal{A}$ is the action space. Every episode starts with a goal selected from $\mathcal{G}$ by the environment and is fixed for the whole episode. We aim to find a goal conditioned policy $\pi : \mathcal{S} \times \mathcal{G} \rightarrow \mathcal{A}$ to maximize the expected cumulative future return $V_{g,\pi}(s_0) = E_\pi[\sum_{t=0}^{\infty} \gamma^t R(s_t, a_t, g)]$, which called goal-conditioned value, or universal value. Universal Value Function Approximators (UVFA) [3] use neural network to model $V(s, g) \approx V_{g,\pi^*}(s)$ where $\pi^*$ is the optimal policy, and apply Bellman equation to train it in a bootstrapping way. Usually, the reward in UMDP is sparse to train the network. For a given goal, the agent can receive non-trivial rewards only when it can reach the goal. This brings a challenge to the learning process.

*Hindsight Experience Replay* (HER) [4] proposes goal-relabeling to train UVFA in sparse reward setting. The key insight of HER is to "turn failure to success", i.e., to make a failed trajectory become success, by replacing the original failed goals with the goals it has achieved. This strategy gives more feedback to the agent and improves the data efficiency for sparse reward environments. Our framework relies on HER to train an accurate low-level policy.

## 4 Universal Goal Reaching

**Problem Definition:** Our universal goal reaching problem refers to a family of UMDP tasks. The state space of our UDMP is a low-dimension manifold in the ambient space. Many useful planning problems in practice are of this kind. Example universal goal reaching environments include labyrinth walking (e.g., AntMaze [31]) and robot arm control (e.g., FetchReach [32]). Their states can only transit in a neighborhood of low-dimensionality constrained by the degree of freedom of actions.

Following the notions in Sec 3, we assume that a goal $g$ in goal space $\mathcal{G}$ which is a subset of the state space $\mathcal{S}$. For example, in a labyrinth walking game with continuous locomotion, the goal can be to reach a specific location in the maze at any velocity. Then, if the state $s$ is a vector consisting of the location and velocity, a convenient way to represent the goal $g$ would be a vector that only contains the dimensions of location, i.e., the goal space is a projection of the state space.

The universal goal reaching problem has a specific transition probability and reward structure. At every time step, the agent moves into a local neighborhood based on the metric in the state space, which might be perturbed by random noise. It also receives some negative penalty (usually a constant, e.g., $-1$ in the experiments) unless it has arrived at the vicinity of the goal. A $0$ reward is received if the goal is reached. To maximize the accumulated reward, the agent has to reach the goal in fewest steps. Usually the only non-trivial reward $0$ appears rarely, and the universal goal reaching problem falls in the category of *sparse reward* environments, which are hard-exploration problems for RL.

**A Graph View:** Assume that a policy $\pi$ takes at most steps $T$ to move from $s$ to $g$ and the reward at each step $r_k$'s absolute value is bounded by $R_{max}$. Let $w_\pi(s, t)$ be the expected total reward along

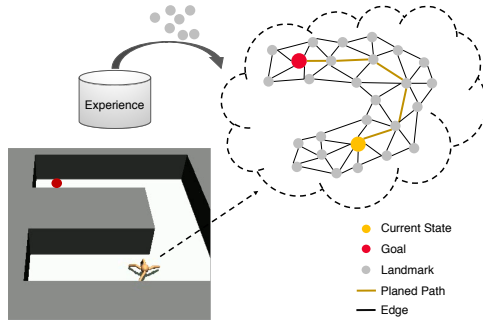

Figure 1: An illustration of our framework. The agent is trying to reach the other side of the maze by planning on a landmark-based map. The landmarks are selected from its past experience, and the edges between the landmarks are formed by a UVFA.

the trajectory, and $d_\pi(s,t) = -w_\pi(s,t)$ for all $s, t$. If $\gamma \approx 1$, we can show that UVFA $V_\pi(s,g)$ can be approximated as (see supplementary for details):

$$V_\pi(s,g) \approx E[w_\pi(s,g)] = E[-d_\pi(s,g)] \tag{1}$$

This suggests us to view the MDP as a directed graph, whose nodes are the state set $\mathcal{S}$, and edges are sampled according to the transition probability in the MDP. The general value iteration for RL problems is exactly the shortest path algorithm in terms of $d_\pi(s,g)$ on this directed graph. Besides, because the nodes form a low-dimensional manifold, nodes that are far away in the state space can only be reached by a long path.

*The MDP of our universal goal reaching problem is a large-scale directed graph whose nodes are in a low-dimensional manifold.* This structure allows us to estimate the all-pair shortest paths accurately by a landmark based coarsening of the graph.

## 5   Approach

In this paper, we choose deep RL algorithms such as DQN and DDPG for discrete and continuous action space, respectively. UVFA [3] is a goal-conditioned extension of the original DQN, while HER (Sec 3), can produce more informative feedback for UVFA learning. Our algorithm is thus based upon HER, and the extension of this approach for other algorithms is also straightforward.

### 5.1   Basic Idea

Our approach aims at addressing the fundamental challenges in UVFA learning. As characterized in the previous section, the UVFA estimation solves a pair-wise shortest path problem, and the underlying graph has a node space of high cardinality. Note that UVFA has to memorize the distance between every state-goal pairs, through trajectory samples from the starting state to the goal, which is much larger than the original state space.

Such large set of state-goal pairs poses the challenge. First, it takes longer time to sample enough state-goal pairs. Particularly, at the early stage, only few state-goal samples have been collected, so learning from them requires heavy extrapolation by networks, which is well known to be unreliable. Second, memorizing all the experiences is too difficult even for large networks.

We propose a map to abstract the visited state space by landmarks and edges to connect them. This abstraction is reasonable due to the underlying structure of our graph — a low-dimensional manifold [33]. We also learn local UVFA networks that only needs to be accurate in the neighborhood of landmarks. As illustrated in Figure 1, an ant robot is put in an "U" Maze to reach a given position. It should learn to model the maze as a small-scale map based on its past experiences.

This solution addresses the challenges. For the UVFA network, it only needs to remember experiences in a local neighborhood. Thus, the training procedure requires much lower sample complexity. The map decomposes a long path into piece-wise short ones, and each of which is from an accurate local network.

Our framework contains three components: a value function approximator trained with hindsight experience replay, a map that is supported by sampled landmarks, and a planner that can find the optimal path with the map. We will introduce them in Sec 5.2, Sec 5.3, and Sec 5.4, respectively.

## 5.2   Learning a Local UVFA with HER

Specifically, we define the following reward function for goal reaching problem:

$$r_t = \mathcal{R}(s_t, a_t, g) = \begin{cases} 0 & |s'_t - g| \leq \delta \\ -1 & otherwise \end{cases}$$

Here $s'_t$ is the next observation after taking action $a_t$. We first learn a UVFA based on HER, which has proven its efficiency for UVFA. In experiments (see Sec 6.3), we find out that the agent trained with HER does master the skill to reach goals of increasing difficulty in a curriculum way. However, the agent can seldom reach the most difficult goals constantly, while the success rate of reaching easier goals remains stable. All these observations prove that HER's value and policy is locally reliable.

One can pre-train the HER agent and then build map for planner. However, as an off-policy algorithm, HER can work with arbitrary exploration policy. Thus we use the planner based on current local HER agent as the exploration policy and train the local HER agent jointly. We sample long horizon trajectories with the planner and store them into the replay buffer. We change the replacement strategy in HER, ensuring that the replaced goals are sampled from the near future within a fixed number of steps to increase the agent's ability to reach nearby goals at the early stage.

The UVFA trained in this step will be used in the planner for two purposes: (1) to estimate the distance between two local states belonging to the same landmark, or between two nearby landmarks; and (2) to decide whether two states are close enough so that we can trust the distance estimation from the network. Although the learned UVFA is imperfect globally, it is enough for the two local usages.

## 5.3   Building a Map by Sampling Landmarks

After training the UVFA, we will obtain a distance estimation $d(s, g)^3$, a policy for any state-goal pair $(s, g)$, and a replay buffer that contains all the past experiences. We will build a landmark-based map to abstract the state space based on the experiences. The pseudo-code for the algorithm is shown in Algorithm 1.

---

**Algorithm 1:** Planning with State-space Mapping (Planner)

---

**Input:** state $obs$, goal $g$, UVFA $Q(s, g, a)$, clip_value $\tau$
**Output:** Next subgoal $g_{next}$
1  Sample transitions $T = (s, a, s')$ from replay buffer $B$
2  $V \leftarrow$ **FPS**$(S = \{s \| (s, a, s') \in T\}) \cup \{g\}$          ▷ Farthest point sampling to find landmarks
3  $W_{ij} \leftarrow \infty$          ▷ Initialize Map as graph $G = \langle V, W \rangle$
4  **for** $\forall (v_i, v_j) \in V \times V$ **do**
5       $w_{ij} \leftarrow \min_a -Q(v_i, v_j, a)$
6       **if** $w_{ij} \leq \tau$ **then**
7           $W_{ij} = w_{ij}$
8  $D \leftarrow$ **Bellman_Ford**(W)          ▷ Calculate pairwise distance
9  $g_{next} \leftarrow \arg\min_{v_i, a} -Q(obs, v_i, a) + D_{v_i, g}$
10 **return** $g_{next}$

---

**Landmark Sampling**   The replay buffer stores visited states during training. Instead of localizing few important states that play a key role in connecting the environment, we seek to sample many states to cover the visited state space.

Limited by computation budget, we first uniformly sample a big set of states from the replay buffer, and then use the farthest point sampling (FPS) algorithm [34] to select landmarks to support the

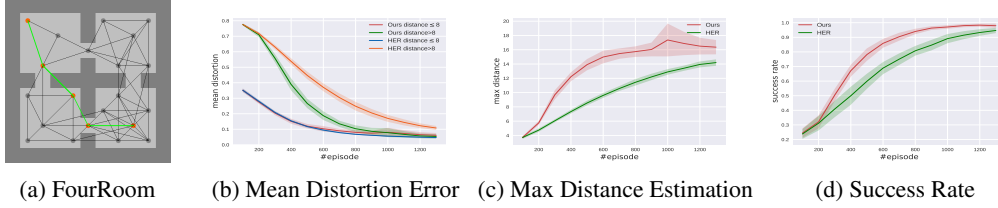

| (a) FourRoom | (b) Mean Distortion Error | (c) Max Distance Estimation | (d) Success Rate |

Figure 2: The results on FourRoom Environment. Figure 2a shows the sampled landmarks and the planned path based on our algorithm. Figure 2c, 2b, 2d are different evaluation metrics of value estimation and success rate to reach the goal.

explored state space. The metric for FPS can either be the Euclidean distance between the original state representation or the pairwise value estimated by the agent.

We compare different sampling strategies in Section 6.3, and demonstrate the advantage of FPS in abstracting the visited state space and exploration.

**Connecting Nearby Landmarks**    We first connect landmarks that have a reliable distance estimation from the UVFA and assign the UVFA-estimated distance between them as the weight of the connecting edge.

Since UVFA is accurate locally but unreliable for long-term future, we choose to only connect nearby landmarks. The UVFA is able to return a distance between any pair $(s, g)$, so we connect the pairs with distance below a preset threshold $\tau$, which should ensure that all the edges are reliable, as well as the whole graph is connected.

With these two steps, we have built a directed weighted graph which can approximate the visited state space. This graph is our map to be used for high-level planning. Such map induces a new environment, where the action is to choose to move to another landmark. The details can be found in Algorithm 1.

### 5.4   Planning with the Map

We can now leverage the map and the local UVFA network to estimate the distance between any state-goal pairs, which induces a reliable policy for the agent to reach the goal.

For a given pair of $(s, g)$, we can plan the optimal path between $(s, g)$ by selecting a serial of landmarks $l_1, \cdots, l_k$, so that the approximated distance will be $\bar{d}(s, g) = \min_{l_1, \dots, l_k} d(s, l_1) + \sum_{i=1}^{k-1} d(l_i, l_{i+1}) + d(l_k, g)$. The policy from $s$ to $g$ can then be approximated as: $\bar{\pi}(s, g) = \pi(s, l_1) + \sum_{i=1}^{k-1} \pi(l_i, l_{i+1}) + \pi(l_k, g)$. Here the summation of $\pi$ is the concatenation of the corresponding action sequence.

In our implementation, we run the shortest path algorithm to solve the above minimization problem. To speed up the pipeline, we first calculate the pairwise distances $d(l_i, g)$ between each landmark $l_i$ and the goal $g$ when episode starts. When the agent is at state $s$, we can choose the next subgoal by finding $g_{next} = \arg\min_{l_i} d(s, l_i) + d(l_i, g)$.

## 6   Experiments

### 6.1   FourRoom: An Illustrative Example

We first demonstrate the merits of our method in the FourRoom environment, where the action space is discrete. The environment is visualized in Figure 2a. There are walls separating the space into four rooms, with narrow openings to connect them. For this discrete environment, we use DQN [35] with HER [4] to learn the Q value. Here, we use the one-hot representation of the x-y position as the input of the network. The initial states and the goals are randomly sampled during training.

We first get $V(s, g)$ from the learned Q-value by equation $V(s, g) = \arg\max_a Q(s, a, g)$, and convert $V(s, g)$ to pairwise distance $D(s, g)$ based on Eq. 1. To evaluate the accuracy of distance estimation, we further calculate the ground truth distance $D_{gt}(s, g)$ by running a shortest path algorithm on

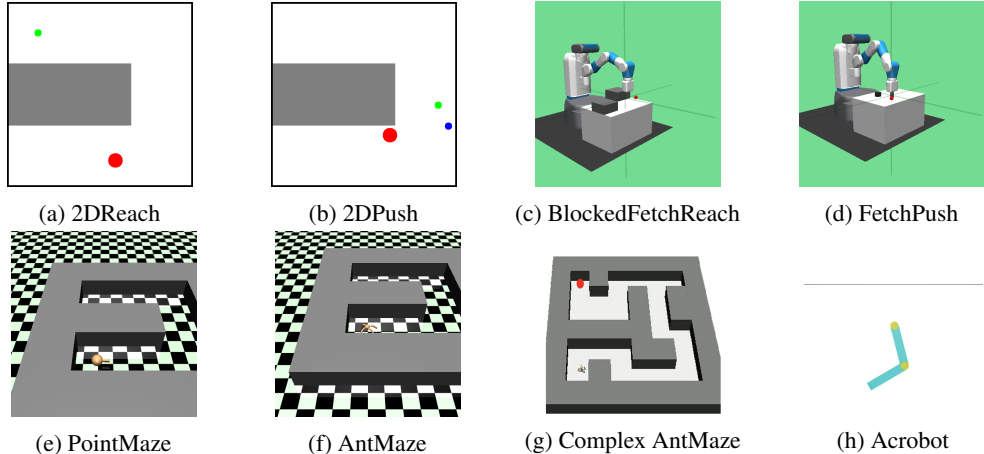

| (a) 2DReach | (b) 2DPush | (c) BlockedFetchReach | (d) FetchPush |

| (e) PointMaze | (f) AntMaze | (g) Complex AntMaze | (h) Acrobot |

Figure 3: The environments we use for continuous control experiments.

the underlying ground-truth graph of maze. Then we adapt the mean distortion error (MDE) as the evaluation metric: $\frac{|D(s,g) - D_{gt}(s,g)|}{D_{gt}(s,g)}$.

Results are shown in Figure 2b. Our method has a much lower MDE at the very beginning stage, which means that the estimated value is more accurate.

To better evaluate our superiority for distant goals, we first convert predicted values to corresponding distances, and then plot the maximal distance during training. From Figure 2c, we can observe that the planning module have a larger output range than DQN. We guess that this comes from the max-operation in the Bellman-Ford equation, which pushes DQN to overestimate the Q value, or in other words, underestimate the distance for distant goals. However, the planner can still use piece-wise correct estimations to approximate the real distance to the goal.

We also compare our method with DQN on success reaching rate, and their performances are shown in Figure 2d. Our method can achieve better accuracy at the early stage.

## 6.2 Continuous Control

In this section, we will compare our method with HER on challenging classic control tasks and MuJoCo [36] goal-reaching environments.

### 6.2.1 Environment Description

We test our algorithms on the following environments:

**2DReach** A green point in a 2D U-maze aims to reach the goal represented by a red point, as shown in Figure 3a. The size of the maze is $15 \times 15$. The state space and the goal space are both in this 2D maze. At each step, the agent can move within $[-1, 1] \times [-1, 1]$ as $\delta_x, \delta_y$ in x and y directions.

**2DPush** The green point A now need to push a blue point B to a given goal (red point) lying in the same U-maze as 2DReach, as shown in Figure 3b. Once A has reached B, B will follow the movement of A. In this environment, the state is a 4-dim vector that contains the location of both A and B.

**BlockedFetchReach & FetchPush** We need to control a gripper to either reach a location in 3d space or push an object in the table to a specific location, as shown in Figure 3c and Figure 3d. Since the original FetchReach implemented in OpenAI gym [37] is very easy to solve, we further add some blocks to increase the difficulty. We call this new environment BlockedFetchReach.

**PointMaze & AntMaze** As shown in Figure 3e and Figure 3f, a point mass or an ant is put in a $12 \times 12$ U-maze. Both agents are trained to reach a random goal from a random location and tested under the most difficult setting to reach the other side of maze within 500 steps. The states of point and ant are 7-dim and 30-dim, including positions and velocities.

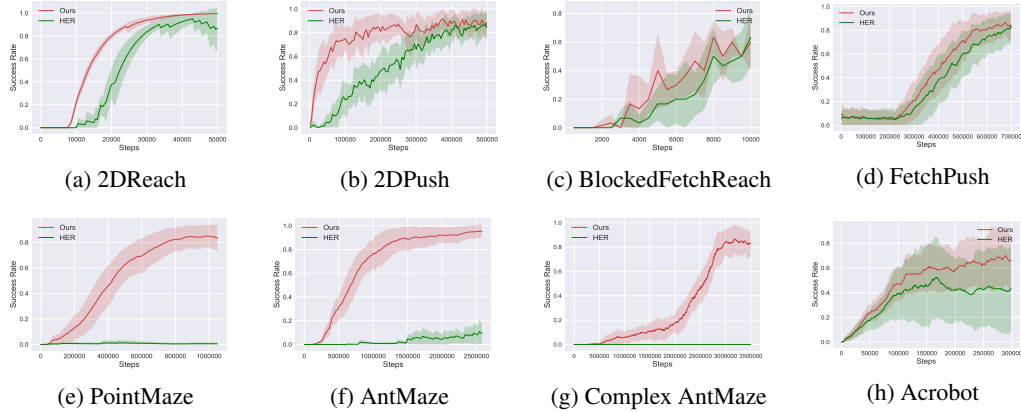

| (a) 2DReach | (b) 2DPush | (c) BlockedFetchReach | (d) FetchPush |
| (e) PointMaze | (f) AntMaze | (g) Complex AntMaze | (h) Acrobot |

Figure 4: Experiments on the continuous control environments. The red curve indicates the performance of our method at different training steps.

**Complex AntMaze** As shown in Figure 3g, an ant is put in a $56 \times 56$ complex maze. It is trained to reach a random goal from a random location and tested under the most difficult setting to reach the farthest goal (indicated as the red point) within 1500 steps.

**Acrobot** As shown in Figure 3h, an acrobot includes two joints and two links. Goals are states that the end-effector is above the black line at specific joint angles and velocities. The states and goals are both 6-dim vectors including joint angles and velocities.

### 6.2.2 Experiment Result

The results compared with HER are shown in Figure 4. Our method trains UVFA with planner and HER. It is evaluated under the test setting, using the model and replay buffer at corresponding training steps.

In the **2DReach** and **2DPush** task (shown in Figure 4b), we can see our method achieves better performance. When incorporating with control tasks, for **BlockedFetchReach** and **FetchPush** environments, the results still show that our performance is better than HER, but the improvement is not so remarkable. We guess this comes from the strict time limit of the two environments, which is only 50. We observe that pure HER can finally learn well, when the task horizon is not very long.

We expect that building maps would be more helpful for long-range goals, which is evidenced in the environments with longer episode length. Here we choose **PointMaze** and **AntMaze** with scale $12 \times 12$. For training, the agent is born at a random position to reach a random goal in the maze. For testing, the agent should reach the other side of the "U-Maze" within 500 steps. For these two environments, the performance of planning is significantly better and remains stable, while HER can hardly learn a reliable policy. Results are shown in Figure 4e and Figure 4f.

We also evaluate our method on classic control, and more complex navigation + locomotion task. Here we choose **Complex Antmaze** and **Acrobot**, and results are shown in Figure 4h and Figure 4g. The advantage over baseline demonstrates our method is applicable to complicated navigation tasks as well as general MDPs.

We also compare our method with Hierarchy RL on AntMaze and our method outperform recent Hierarchy RL methods. See supplementary material for details.

### 6.3 Ablation Study

We study some key factors that affect our algorithm on AntMaze.

**Choice of clip range and landmarks** There are two main hyper-parameters for the planner – the number of landmarks and the edge clipping threshold $\tau$. Figure 6a shows the evaluation result of the model trained after 0.8M steps in AntMaze. We see that our method is generally robust under different choices of hyper-parameters. Here $\tau$ is the negative distance between landmarks. If it's too

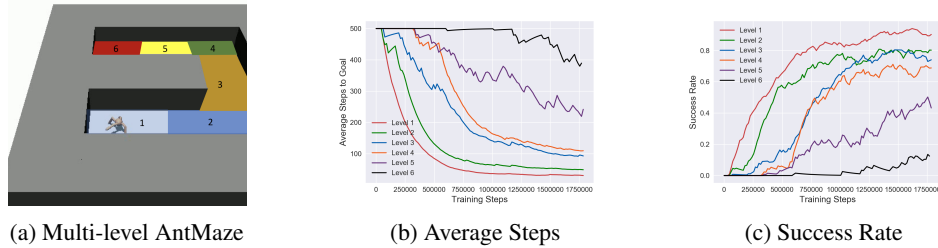

(a) Multi-level AntMaze        (b) Average Steps        (c) Success Rate

Figure 5: AntMaze of multi-level difficulty. Figure 5b and Figure 5c is the average steps and success rate to reach different level of goals, respectively.

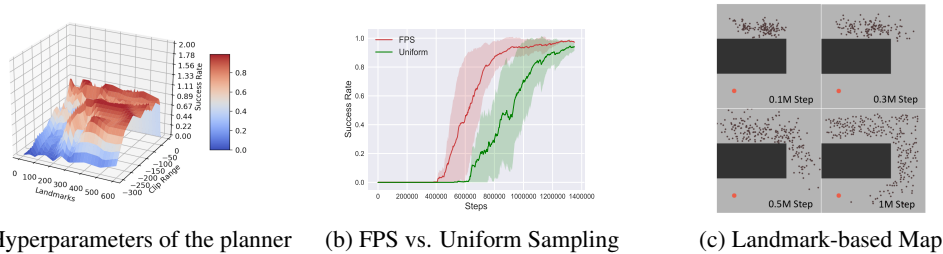

(a) Hyperparameters of the planner    (b) FPS vs. Uniform Sampling    (c) Landmark-based Map

Figure 6: Figure 6a shows the relationship with the landmarks and clip range in the planner. Figure 6b shows FPS outperforms uniform sampling. And Figure 6c is the landmark-based map at different training steps constructed by FPS.

small, the landmarks will be isolated and can't form a connected graph. The same problem comes when the landmarks are not enough.

**The local accuracy of HER** We evaluate our model trained between 0~2.5M steps, for goals of different difficulties. We manually define the difficulty level of goals, as shown in Figure 5a. Goal's difficulty increases from Level 1 to Level 6. We plot the success rate as well as the average steps to reach these goals. We find out that, for the easier goals, the agent takes less time and less steps to master the skill. The success rate and average steps also remain more stable during the training process, indicating that our base model is more reliable and stable in the local area.

**Landmark sampling strategy comparison** Our landmarks are dynamically sampled from the replay buffer by iterative FPS algorithm using distances estimated by UVFA, and get updated at the beginning of every episode. The FPS sampling tends to find states at the boundary of the visited space, which implicitly helps exploration. We test FPS and uniform sampling in fix-start AntMaze (The ant is born at a fixed position to reach the other side of maze for both training and testing). Figure 6b shows that FPS has much higher success rate than uniform sampling. Figure 6c shows landmark-based graph at four training stages. Through FPS, landmarks expand gradually towards the goal (red dot), even if it only covers a small proportion of states at the beginning.

## 7 Conclusion

Learning a structured model and combining it with RL algorithms are important for reasoning and planning over long horizons. We propose a sample-based method to dynamically map the visited state space and demonstrate its empirical advantage in routing and exploration in several challenging RL tasks. Experimentally we showed that this approach can solve long-range goal reaching problems better than model-free methods and hierarchical RL methods, for a number of challenging games, even if the goal-conditioned model is only locally accurate. However, our method also has limitations. First, we empirically observe that some parameters, particularly the threshold to check whether we have reached the vicinity of a goal, needs hand-tuning. Secondly, a good state embedding is still important for the learning efficiency of our approach, since we do not include heavy component of learning state embedding. Thirdly, we find that in some environments whose intrinsic dimension is very high, especially when the topological structure is hard to abstract, sample-based method is not enough to represent the visited state space. And for those environments which is hard to obtain a reliable and generalizable local policy, this approach will also suffer from the accumulated error.

## Footnotes

[2] $s$ is the current state and $g$ is the goal.

[3]If the algorithm returns a $Q$ function, we will calculate the value by selecting the optimal action and calculate the $Q$ function and convert to $d$ by Eq. 1

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
