[Supplementary Material · supple_2.pdf]

# Supplementary Material for *Mapping State Space using Landmarks for Universal Goal Reaching*

## 1 Overview

This material contains five parts, in addition to the illustrations and experiments in the main paper. The first part contains a proof about the feasibility of approximating value by pairwise distance, which is introduced in Section 4 of the main paper. The second part is a Pseudo-code of our framework, which clearly shows the algorithm. In the third part, we compare our method with the state-of-the-art hierarchical reinforcement learning algorithms, HIRO [1] and HAC [2] on the AntMaze of size $24 \times 24$. And we also show the forgetting issue of HER [3], as a result of the limited network capacity. In the last part, we show the implementation parameters for experiments.

## 2 Proof of the Approximation in Section 4

In Section 4 of the main paper, we propose to view the MDP as a directed graph, and this is true when $\gamma \approx 1$.

Assume that a policy $\pi$ takes at most steps $T$ to move from $s$ to $g$ and the reward at each step $r_k$'s absolute value is bounded by $R_{max}$. Let $w_\pi(s,t)$ be the expected total reward along the trajectory, and $d_\pi(s,t) = -w_\pi(s,t)$ for all $s,t$. If $\gamma = 1 - \epsilon$, we can prove that [1]: $|V_\pi(s,g) - w_\pi(s,g)| \leq T^2 R_{max}\epsilon$. Thus, when $\gamma \approx 1$ and $T^2 R_{max}(1-\gamma) \xrightarrow{+} 0$, UVFA can be approximated as:

$$V_\pi(s,g) \approx E[w_\pi(s,g)] = E[-d_\pi(s,g)] \tag{1}$$

In this case, it is easy to show that the value iteration based on Bellman Equation $V_{\pi^*}(s,g) = R(s,a,g) + \gamma\mathbb{E}[V_{\pi^*}(s',g)]|_{s'\sim\mathcal{P}_{\pi^*}(\cdot|s,a)}$ implies $w_{\pi^*}(s,g) \approx R(s,a,g) + w_{\pi^*}(s',g)|_{s'\sim\mathcal{P}_{\pi^*}(\cdot|s,a)}$, where $\mathcal{P}_{\pi^*}$ is the transition probability of optimal policy $\pi^*$.

To prove this, note that $\gamma$ is smaller than 1, we can further replace $\gamma$ with $\epsilon = 1 - \gamma$. When $\epsilon \xrightarrow{+} 0$, we can approximate $(1-\epsilon)^k$ by its first-order Taylor expansion $1 - \epsilon k$. Thus we have:

$$
\begin{aligned}
|V_\pi(s,g) - w_\pi(s,g)| &= |E[\sum_{k=1}^{T} r_k\gamma^{k-1}] - E[\sum_{k=1}^{T} r_k]| \\
&= |E[\sum_{k=1}^{T} r_k(1-\epsilon)^{k-1}] - E[\sum_{k=1}^{T} r_k]| \\
&\approx |E[\sum_{k=1}^{T} r_k - (k-1)\epsilon r_k] - E[\sum_{k=1}^{T} r_k]| \\
&= |\sum_{k=1}^{T} (k-1)\epsilon E[r_k]| \\
&\leq T^2 R_{max}\epsilon \xrightarrow{+} 0,
\end{aligned}
$$

|             | 0.5M | 0.75M | 1M   | 1.25M | 1.5M | 1.75M | 2M  |
|-------------|------|-------|------|-------|------|-------|-----|
| Ours Sparse | 0.0  | 0.03  | 0.3  | 0.4   | 0.45 | 0.5   | 0.5 |
| HIRO Sparse | 0.0  | 0.0   | 0.0  | 0.0   | 0.0  | 0.0   | 0.0 |
| Ours Dense  | 0.0  | **0.09** | **0.45** | **0.5** | **0.7** | **0.8** | **0.9** |
| HIRO Dense  | 0.0  | 0.0   | 0.0  | 0.1   | 0.4  | 0.6   | 0.8 |

Table 1: Success Rate on Large AntMaze at different training steps.

So we finish the proof of the relationship between $V_\pi$ and $w_\pi$ when $\gamma \to 1$, as mentioned in the main paper.

## 3  Training Algorithm Outline

The Pseudo-code for the training algorithm is listed in Algorithm **??**. The latter is how we build the map and select subgoals by planning in a dynamic programming way.

---
**Algorithm 1** Train and Test with Planning

---
**Input:** current observation $obs$, desired goal $g$
**for** *every training step* **do**
    with probability $\alpha$:
      action = Actor($obs, g$) + noise
    with probability $1 - \alpha$:
      action = Planner($obs, g$)
    next_obs = env.step(action)
    Train actor and critic network with **hindsight experience replay**
    store trajectories in replay buffer when episode ends
**end**
**for** *every test step* **do**
    action = Planner($obs, g$)
    next_obs = env.step(action)
**end**

---

## 4  Comparison with HRL

We compare our method with HRL algorithms on large AntMaze (size $24 \times 24$), as shown in Table 1. We choose to compare with HIRO [1], which is the SOTA HRL algorithm on AntMaze, and HAC [2], which also uses the hindsight experience replay. We test these algorithms with the published codes[2][3], under both sparse reward setting and dense reward setting.

On sparse reward setting, our algorithm can work well and reach the goal at the very early stage (*Ours sparse* in Table 1). In contrast, neither HAC nor HIRO are able to reach the goal in 2M steps. HIRO doesn't use HER to replace the unachievable goals, which makes such setting very challenging for the algorithm.

For dense reward setting, the map planner can obtain a high success rate at very early stage shown as *Ours dense* in Table 1. Compared with *HIRO dense*, we can see that a planner can reach distant goals sooner, since we don't need to train a high-level policy to propose subgoals for the low-level agent.

HAC introduced several complex hyper-parameters, and we couldn't make it work well for both settings.

## 5  The Forgetting Issue of HER

We observe that HER may forget how to reach the ultimate goal even if it learns to reach it some steps ago. For AntMaze, as shown in the main paper, the success rate for pure HER is always below

Figure 1: The success rate of HER to reach a random goal after we flip the training and testing setting.

0.2. Since the goal for testing is the most difficult one, to better evaluate this issue for a larger goal space $\mathcal{G}$ of different difficulties, we then flip the setting for training and testing, i.e., for training, the agent aims to reach a fixed goal at the other side of the maze, but for testing, the agent is born at a random location and tries to reach a random goal. Here we use a well-pretrained model, which has almost 0.7 success rate to reach a random sampled goal within 200 steps. We then retrain it to reach a fixed goal under the new setting. We observe that, although its performance to reach a fixed goal is slowly increasing, its ability to reach a randomly picked goal in the maze drops to $0.5 \sim 0.6$.

# 6 Implementation Parameters

We use the following DDPG architecture and hyper-parameters for all the experiments:

**Q/Critic Network Layers**: 5

**Q/Critic Network Hidden Dimension**: 400

**Policy Network Layers**: 3

**Policy Network Hidden Dimension**: 400

**Network Activation**: ReLU

**Noise**: For the environments except AntMaze, we use OU-noise with $\sigma = 0.02$ for DDPG. For AntMaze, we use 0.2 epsilon-greedy to improve exploration.

**Discount Factor**: 0.99

**Batch Size**: 128

**Actor Learning Rate**: 0.0003

**Critic Learning Rate**: 0.0003

**Target Network Update Ratio**: 0.005

**HER Replace Ratio**: 0.8

**Episode Length**: 500 for PointMaze and AntMaze; 50 for the Fetch/Push environments; 1500 for Complex AntMaze; 200 for Acrobot.

**Distance Threshold** The distance threshold $\delta$ is used to judge whether a goal has been reached in HER module. This is different for each environment:

1. **2DMaze & 2DPush & Acrobot**: 0.03

2. **FetchPush & FetchReach**: 0.025

3. **PointMaze & AntMaze & Complex AntMaze**: 0.1

## Footnotes

[1] When $\epsilon \xrightarrow{+} 0$, we can approximate $(1-\epsilon)^k$ by its first-order Taylor expansion $1 - \epsilon k$.

[2]HIRO: https://github.com/tensorflow/models/tree/master/research/efficient-hrl

[3]HAC:https://github.com/andrew-j-levy/Hierarchical-Actor-Critc-HAC-