[Reviews · NeurIPS 2019]

Reviewer 1



The paper presents a semi-parameteric model for long-term planning in a general space of problems. It works by training parametric goal-conditioned policies accurate only on local distances (i.e. when current state and goal are within some distance threshold) and leveraging the replay buffer to non-parametrically sample a graph of landmarks which the local goal-conditioned policy can accurately produce paths between. Moving to any goal state is then accomplished by (1) moving to the closest landmark using the goal-conditioned policy, (2) planning a path to the landmark closest to the goal using value-iteration on the (low-dimensional) graph of landmarks, (3) using the goal-conditioned policy to get to the goal state from the closest landmark. The paper essentially tackles the problem that goal-conditioned policies, or Universal Value Function Approximators (UVFA), degrade substantially in performance as the planning horizon increases. By leveraging the replay buffer to provide way-points for the algorithm to plan locally along, accuracy over longer ranges is maintained. The method is close to a heuristic in the sense that there is no learning in this higher-level graph structure, it is largely agent designer prior knowledge to get UVFAs working at longer ranges. Despite this, I still think the method can be impactful as a first step to solving this challenging long-range planning problem. The evaluation settings demonstrate that the method is viable and improves upon baselines for a variety of continuous control experiments. Unfortunately, most of the improvements are in navigation domains, and in this case I believe a very similar method for navigation has already been done (Semi-parametric Topological Memory [1], which I believe should be cited given its close similarity to this method). Therefore a more informative evaluation setting would be in tasks further away from navigation, and more towards general MDPs which is where I think this paper's contribution is. For example, seeing this implemented on Atari games would be a far more interesting result. It would also be great seeing this method used to drive frontier exploration (doing random exploration near states which have been visited the least number of times). In conclusion, I believe the paper's idea of leveraging the replay buffer to build a set of non-parametric waypoints is an impactful contribution. Unfortunately, a relatively uninformative evaluation setting with too much focus on navigation (bringing it a bit too close to previous work [1]), makes me suggest a score of 6. Some other points: - Figure text is extremely small and difficult to read. - It is hard to contextualize Figure 4(b) without knowing the lower-bound on average steps to goal. It would be easier to read this graph if you added the lower-bound as a dotted line. - How are the curves in Figure 3 generated? Are these the same HER model but in one evaluation you use planning and one you use the HER policy directly? Or is the red curve also using the planner at training time? - Also, what replay buffer do you use at each point along the curves? The one encompassing all states up to that step? Or the replay buffer at the end of training? - I believe the word "unanimous" on line 22 means things are in agreement, which doesn't seem to fit when describing tasks. [1] Savinov, Nikolay, Alexey Dosovitskiy, and Vladlen Koltun. "Semi-parametric topological memory for navigation." arXiv preprint arXiv:1803.00653 (2018).

Reviewer 2



From what I understand, the goal g is typically a projection of the state s of the agent. The agent could be in different states s_1 and s_2 at the same goal g, but states s_1 and s_2 could be not easily reachable one from the other. For example, g could be a location on manifold (e.g., a city map) and s_1 and s_2 could be two different states (same position but opposite orientations) of a mobile agent (e.g., a car). To go from s_1 to s_2 is not trivial as it requires changing the orientation of the car (it might imply around a block. If the information that is used for planning corresponds to only pairs of (s, g), would not the difference between states at goal locations be lost? In that case, can we still write that the cumulated rewards for a complex task d(s, g) is the sum of d(l_k, l_{k+1}) for consecutive landmarks? Or does it simply mean that g should not be only a position, but also an orientation, or alternatively, that the planner works on pairs of states, rather than pairs of state-goals? Minor comments: In section 5.1, explain why the cardinality of the state-goal pairs is O(R^{2d}). The color code on Figure 4 could be consistent between (a) and (b) / (c).

Reviewer 3



This paper proposes to adopt state abstraction method to improve sample efficiency of learning the universal value function approximator (UVFA) for long-range goals. The key idea is building the high-level graph of given MDP by sampling the landmark states, which is uniformly sampled from the state space, and estimating the connectivity (edge) between them. In high-level, the transition between landmark states can be handled by planning on the high-level graph, while the local transition (i.e., current state to near-by landmark or transition between neighboring landmarks) can be handled by UVFA. The proposed method has been evaluated on Mujoco domain including navigation tasks, and significantly outperformed the baseline method: DDPG+HER. Overall, the paper is easy to follow. However, the main limitation of this paper is the lack of contribution. The proposed method can be seen as a relatively straightforward combination of existing works: UVFA+HER+state abstraction and planning [1], which are missing in the related work section. The idea of building the high-level graph based on the sampled landmark states, and performing the planning (i.e., value iteration) in the graph to find the shortest path to the goal has been already proposed in [1]. The salient differences from the existing works are the landmark sampling, and using the value function as the distance metric between landmark, which is a relatively minor technical details. The followings are the related works that could be included in the paper. [1] Zhang, Amy, et al. "Composable Planning with Attributes." International Conference on Machine Learning. 2018. [2] Folqué, David, et al. "Planning with Arithmetic and Geometric Attributes." arXiv 2018. [3] Liu, Evan Zheran, et al. "Learning Abstract Models for Long-Horizon Exploration." 2018. Regarding the landmark sampling step, the rebuttal addressed this issue quite well. I suggest authors to make this point more clear in the paper. I'm increasing my score to 5. Lastly, the experiment section is relatively light. The author might want to include other baselines such as HIRO [28] (which is already cited in the paper). Minor comments: - The texts in the graphs are too small when printed out. - In section 6.2.1, “planning” paragraph, what is the actor network in DQN? - In section 6.3, a line break is missing before “Choice of clip range and landmarks” paragraph.

[Author Response · NeurIPS 2019]

We thank all reviewers for the constructive comments. We will carefully proofread the paper and add suggested
references. Some suggested papers are from unpublished arXiv papers ([2,3] suggested in R3's review). According to
the review policy, reviews should not be based on them, but nonetheless, we will still comment them for clarification.

How to learn a structured model for model-based RL is a hot topic. We attempt to solve goal-conditioned policies in
large MDPs with *sparse* rewards, in which exploration and routing across remote states are both extremely challenging.
Our approach maintains a *dynamic landmark-based map* abstracting the *visited state space* to facilitate the routing
and exploration. To build the map, selecting landmarks and adding edges among them are both based on *learned*
information of the environment. Experimentally we showed that the approach can solve long-range goal reaching
problems better than model-free methods and hierarchical RL methods, for a number of challenging games.

**Significant Concerns:**

**Novelty in Map Construction and Replay Buffer Usage (R1&R3).** Instead of trivially mapping the whole state
space by uniform sampling (R3's review), our landmarks are dynamically sampled from
the replay buffer by iterative FPS algorithm using distances estimated by the UVFA
$Q$-function (LN203-204) and get updated at the beginning of every episode. The FPS
sampling tends to find states at the boundary of the visited space, which implicitly helps
exploration (LN50-53). Figure on the right shows selected landmarks at four training
stages on AntMaze with a fixed starting point at the upper-left corner. Landmarks expand
gradually towards the goal (red dot), even if it only covers a small proportion of states at
the beginning. FPS has much higher success rate than uniform sampling: 0.4 vs 0.15 at
0.5M training step, 0.7 vs 0.3 at 0.75M training step, and 0.95 vs 0.7 at 1M training step.
*This map construction strategy is adaptive to the capability of the policy network, the*
*UVFA network, and the task (rewards), without assuming domain-specific knowledge.*

**Comparison with [1,2,3] suggested by R3.** [1] and [2] (unpublished) learned environment models using domain
knowledge with explicit supervision. They required user-defined attributes and learned the state-attribute mapping in a
supervised way. [3] (unpublished) proposed a complex system of relevant high-level idea with many hyper-parameters
(19) for very different problem settings and solutions. According to the review policy, discussions are left in future work.

**No Comparison with HRL (R3).** Sec 4 of supplementary compared our method with SOTA HRL algorithms (HIRO
and HAC with authors' published code) on big AntMaze of size ($24 \times 24$). The game is quite challenging for mixing
control (30D) and navigation (2D) in states. With sparse rewards, neither HIRO nor HAC can solve it, but ours can
solve it efficiently. With dense rewards, our method outperforms HIRO and HAC significantly (recent results on HAC).

**Applicability to General MDPs (R1 & R2).** Our framework is theoretically applicable to general MDPs. Indeed, the
state space of a few MDPs in our experiments mixes information NOT about location for navigation: AntMaze and
PointMaze use 28D and 4D for the control of robot body, FetchReach uses 7D for the control of gripper, and FetchPush
is a robot-object interaction task that has 15D for the object under manipulation and 7D for gripper control. Note
that we do not use any information (such as control cost) other than the sparse rewards for goal reaching. This set of
environments have basically covered the commonly used games for goal-conditioned MDPs.

We also recently tested on a non-navigational control task (Acrobot), an Atari game (MonteZuma's Revenge), and a
complex locomotion+navigation task (complex AntMaze). Note that the original Acrobot and
MonteZuma's Revenge are not universal goal-reaching tasks, but we re-wrap them by sampling
goals. For Acrobot, goals are states that the end-effector is above a certain line at specific joint
angles and velocities. For MonteZuma's Revenge, goals are sampled according to the RND-based
novelty value. *The modification makes them much harder than the original version for demanding*
*the ability to reach arbitrary goals.* One Acrobot, our method achieved a higher success rate
than HER (0.65 vs 0.25 after 0.5M steps). On MonteZuma's Revenge, our method outperformed

HER+RND by a large margin and can successfully reach the key in the first room while HER fails.
Note that our backbone algorithm, DQN, is well-known to perform poorly on this game. Given
more time, we would explore other favorable backbone algorithms, like PPO. Finally, we evaluated our method on a
larger and more complex AntMaze, which requires 1500 steps (HIRO and HAC both fail). Our method can occasionally
achieve the goal at an early stage ($\sim$0.5M training steps), and converged at 0.85 success rate before 3M steps.

**Individual Concerns:**

**Figure 3 and Figure 4(b) (R1).** For Figure 3, our method will also use the planner to collect data at training time. The
curve shows the evaluation performance. For Figure 4(b), it shows that a harder goal requires more steps to reach, and
the performance of HER is more unstable.

**State Space and Goal Space (R2).** Usually it is hard and unnecessary to precisely realize intermediate subgoals for
long-term planning. In AntMaze, the landmarks include only the target position but not any locomotion information,
and we found that this is enough for the long-term planning. However, for other environments like the 2DPush and
FetchPush, the goal vector only describes the target objects' location but loses information of the gripper, which is
essential for planning. So in these environments, the landmarks are selected in the original state space. A second HER
from the last landmark to the goal (the object) is learned to complete the planning.

**Actor Network and DQN (R3).** For DQN, the action is generated from the maximal Q-value, which is different from
DDPG. But the solutions of using Q-value to connect landmarks are similar.

[Meta-Review · NeurIPS 2019]

Reviewers liked the approach of combining local navigation using a UVFA trained by HER with global planning based on shortest paths in a graph constructed from a buffer of landmarks. At the same time, reviewers had some concerns regarding clarity of presentation, the similarity of the proposed approach to existing work (namely “Semi-parametric topological memory for navigation” by Savinov et al.), as well as how specific the proposed algorithm is to navigation problems. The rebuttal provided additional experiments on several new domains and included additional discussion of related work, leading one reviewer to raise the score. In the end the ACs found this work to be sufficiently new and promising to warrant acceptance, but we ask the authors to 1) address the concerns regarding related work including Savinov et al. and 2) include a clearer statement of the full approach (an algorithm block would be great) in the camera ready version.